# A Simple Analysis Method of Specific Anammox Activity Using a Respirometer

**Jaecheul Yu** [1,2]🄳**, Yeonju Kim** [3]🄳**, Jeongmi Kim** [2]**, Soyeon Jeong** [2]**, Seongjae Park** [2] **and Taeho Lee** [2,*]🄳

1 Institute for Environment and Energy, Pusan National University, Busan 46241, Korea; yjcall0715@pusan.ac.kr
2 Department of Civil and Environmental Engineering, Pusan National University, Busan 46241, Korea; magicstar2@nate.com (J.K.); jeongsy002@gmail.com (S.J.); sjker25@naver.com (S.P.)
3 Disaster Scientific Investigation Division, National Disaster Management Research Institute, Ulsan 44538, Korea; kyj9219@gmail.com
* Correspondence: leeth55@pusan.ac.kr

**Abstract:** Anaerobic ammonium oxidation (anammox) is a biological nitrogen removal process with attractive prospects, such as no carbon addition, less aeration, lower greenhouse gas generation, and lower sludge production. However, it is difficult to maintain a stable anammox process since the anammox bacteria have a slow growth rate and high sensitivity to many factors. Therefore, it is very important to analyze and maintain the anammox activity as a process indicator for its successful operation. The conventional method for measuring the concentration of nitrogen compounds, such as ammonium, nitrite, or nitrogen gas is inconvenient during the reaction time for specific anammox activity (SAA) analysis, which can result in an inaccurately determined SAA due to the substrate loss and temperature change. In this study, a respirometer was utilized to analyze the SAA. The SAA values from a respirometer (rSAA) showed a similar pattern to the SAA values (mSAA) from the conventional method. All of the SAA analyses showed the highest value at 35 °C with a granule size of <1 mm. Statistical analysis showed no significant differences regardless of the analysis method, since the *p*-values for the *t*-test and Wilcoxon rank-sum test were >0.05. Therefore, the respirometer can be used as a simple and efficient tool for SAA analysis. Moreover, the operating maintenance and management of the anammox process can be improved due to the simple SAA analysis in the field.

**Keywords:** anammox; granule; respirometer; SAA analysis; statistical analysis





## 1. Introduction

The anaerobic ammonium oxidation (anammox) process has several advantages over the conventional nitrification–denitrification process, such as no carbon addition requirements, less aeration, lower greenhouse gas generation, and lower sludge production [1,2]. However, a stable performance is difficult to maintain since the anammox bacteria have a slow growth rate and are highly sensitive to environmental conditions, such as substrate concentration, granule size, organic matter, temperature, etc. [1,3]. Therefore, it is essential to maintain a sufficient amount of anammox biomass in the reactors for successful operation. As a result, measurements of the anammox activity can indicate the parameters of the stable process operation since they identify the inhibitory effects on anammox bacteria from various factors.

In general, the anammox activity is expressed as a specific anammox activity (SAA). It can be determined based on the amount of ammonium ($gNH_4^+/gVSS/day$) and nitrite ($gNO_2^-/gVSS/day$) consumed or the amount of nitrogen gas produced ($gN_2/gVSS/day$) per biomass concentration over the reaction time, as shown in the following stoichiometry relationship (Equation (1)) [4]. Numerous studies analyzed the SAA to examine the effects of various environmental factors, such as zinc [5], sludge type [6], temperature, and pH [5] or salt [7] on the anammox process. However, measuring the concentrations of nitrogen compounds, such as ammonium, nitrite or nitrogen gas is inconvenient during the reaction

time for the SAA analysis, which can result in an inaccurately determined SAA due to the substrate loss and temperature change.

$$NH_4^+ + 1.32NO_2^- + 0.066HCO_3^- + 0.13H^+ \rightarrow 1.02N_2 + 0.26NO_3^- + 0.066CH_2O_{0.5}N_{0.15} + 2.03H_2O \qquad (1)$$

The respirometer is a device that analyzes the microbial activity simply and rapidly by in situ measurements of the gas consumed or produced from the respiration of microbes [8]. Numerous studies have made use of respirometers to evaluate the activity of iron-and sulfide-oxidizing bacteria [9,10], sulfur-oxidizing bacteria [11], thermophilic bioleaching archaea [12], hydrogen-producing bacteria [13], soil bacteria [14], activated sludge [15,16], anaerobic sludge [17], marine bacteria [18], and sediment [19]. These results can be used for wastewater treatment operation and management [20]. In addition, SAA can be measured using a respirometer since anammox bacteria produce nitrogen gas from ammonium and nitrite. However, to the best of our knowledge, there are little reports of SAA analysis using a respirometer.

This study evaluated a simple method for the determination of SAA using a respirometer and applied it to investigate the effects of granule size and temperature on the SAA. These results were compared with the SAA results of conventional analysis methods, providing readers with comprehensive and useful reference information for SAA analysis using a respirometer.

## 2. Materials and Methods

### 2.1. Anammox Granules

Anammox granules of the laboratory-scale anammox cultivation reactor (working volume 36 L), which has been operating with 200 mg$NH_4^+$-N and 200 mg$NO_2^-$-N/L were used in this study. This reactor was operated in a continuous stirred mode under 8–10 h of HRT. A nitrogen removal efficiency of approximately 85% and nitrogen removal rate of 0.82–1.02 kg/m$^3$/day were achieved. Moreover, the granules were enriched with uncultured anammox bacterium (68.1%), Ca. *Brocadia* (8.6%), Ca. *Jettenia* (1.9%), and Ca. *Kuenenia* (2.6%).

### 2.2. Batch Test

A batch test was performed with a working volume of 36 L and was conducted in two steps. In step 1, the granules that were not classified by size were used. Additionally, gas production was measured until all of the reactions were completed to determine the optimal reaction time for SAA analysis using the respirometer (BRS-100, EETech Co., Chuncheon, Korea) (Figure 1). In step 2, the experiments were conducted according to the granule size (<1, 1–2, and >2 mm) and temperature (30, 35, and 40 °C). Moreover, the granules were separated using a sieve for each size.

Herein, N$_2$ gas production was calculated using two methods based on the actual measured gas ($\Delta_r N_2$) using a respirometer (Equation (2)). In addition, theoretical gas production ($\Delta_m N_2$) was calculated from the actual consumed ammonium and nitrite concentrations based on the following stoichiometry equation (Equations (3) and (4)):

$$\Delta_r N_2(molN_2/min) = \frac{P \cdot \Delta N}{R \cdot T \cdot \Delta t} \qquad (2)$$

$$\Delta_m N_2(molN_2/min) = \frac{1.02 \cdot (n\Delta A) \cdot V_L}{M \cdot \Delta t} \text{ at } \frac{the \ removed \ NO_2^- - N}{the \ removed \ NH_4^+ - N} \geq 1.32 \qquad (3)$$

$$\Delta_m N_2(molN_2/min) = \frac{1.02 \cdot (n\Delta Ni) \cdot V_L}{M \cdot \Delta t} \text{ at } \frac{the \ removed \ NO_2^- - N}{the \ removed \ NH_4^+ - N} < 1.32 \qquad (4)$$

where $\Delta N$ is the net increase in N$_2$ gas volume (L) measured from a respirometer over the reaction time; $\Delta t$ is the reaction time (min); $R$ is the ideal gas coefficient (0.082 atm L/mol·K); $T$ is the temperature (K); $\Delta A$ is $NH_4^+$-N removed (g/L) during the reaction time; $P$ is pressure (1 atm); $\Delta Ni$ is $NO_2^-$-N removed (g/L) during the reaction time; $P$ is pressure

(1 atm); $V_L$ is the volume of liquid (0.1 L); M is the molecular weight of nitrogen gas (28 g/mol); and $n$ is the $N_2$ gas modifying factor based on the stoichiometry equation (Equation (1)), which is one when the ratio of removed $NO_2^--N$ divided by the removed $NH_4^+-N$ is $\geq 1.32$, and 1.32 when the ratio of removed $NO_2^--N$ divided by the removed $NH_4^+-N$ is <1.32.

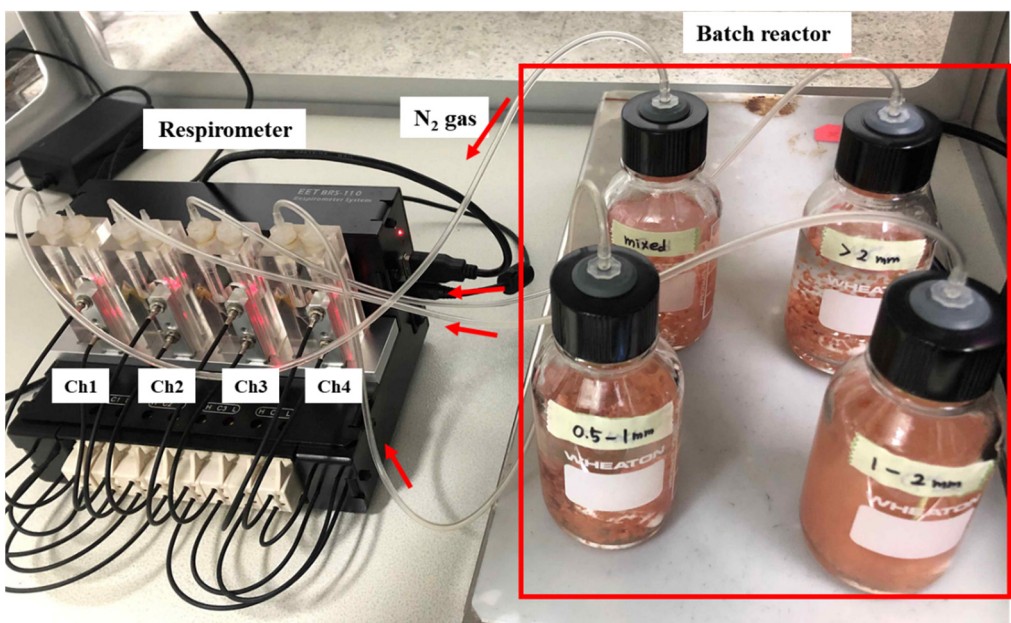

**Figure 1.** Batch test for SAA analysis using a respirometer.

SAA was calculated from the $N_2$ gas production rate divided by the biomass concentrations (gVSS/L) in the vial (Equations (5) and (6)):

$$\text{rSAA } (gN_2/gVSS/day) = \frac{\Delta rN_2 \cdot M \cdot f}{XV_L} \tag{5}$$

$$\text{mSAA } (gN_2/gVSS/day) = \frac{\Delta mN_2 \cdot M \cdot f}{XV_L} \tag{6}$$

where rSAA is the SAA calculated based on the $N_2$ gas production rate using a respirometer; mSAA is the SAA calculated based on the $N_2$ gas production rate from the actual removed ammonium and nitrite; X is the biomass concentration (gVSS/L); M is the molecular weight of nitrogen gas (28 g/mol); $f$ is the time modifying factor (1440 min/day); and $V_L$ is the volume of liquid (0.1 L).

All of the batch experiments (3 gVSS/L) were performed in duplicate under a thermostatic chamber maintained at 35 °C, except for step 2 (On-lab Co., Busan, Korea). The reactors were carried out under a complete mixing condition using a stirrer at 120 rpm. Before the experiments, each reactor was purged with argon gas for 10 min to remove the oxygen and ensure anaerobic conditions in the reactor. The composition of the medium was as follows: 100 mg/L $NH_4^+-N$, 100 mg/L $NO_2^--N$, 1.010 g/L $NaHCO_3$, 0.055 g/L $KH_2PO4$, 0.005 g/L $CaCl_2 \cdot 5H_2O$, 0.5 g/L $MgSO_4 \cdot 7H_2O$, 0.010 g/L $FeSO_4 \cdot 7H_2O$, 0.005 g/L EDTA.

*2.3. Analysis*

The biomass concentration (gVSS/L) was analyzed according to the standard method (APHA, 2005). All of the liquid samples from each vial were filtered through 0.22 μm disposable filters (RephiQuik Syringe Filter, RephiLe Bioscience Ltd., Shanghai, China) and stored in microcentrifuge tubes prior to further analysis. Ammonium and nitrite were analyzed using a kit (Humas Co. Ltd., Daejeon, Korea) according to the standard method (APHA, 2005).

Statistical analysis was performed using R [21]. The significance of the differences between rSAA and mSAA was evaluated by comparing the $p$-values of the $t$-test and Wilcoxon rank-sum test following the Shapiro–Wilk normality test ($p$-value < 0.05).

## 3. Results and Discussion

### 3.1. Changes in $N_2$ Gas Production and rSAA According to the Reaction Time

An investigation of $N_2$ gas production was conducted until the end of the reaction to determine the saturation point for SAA using a respirometer for the measurement of $N_2$ gas production (Figure 2). In this case, the accumulated $N_2$ gas production increased until 360 min and remained unchanged until the end of the reaction (saturated point).

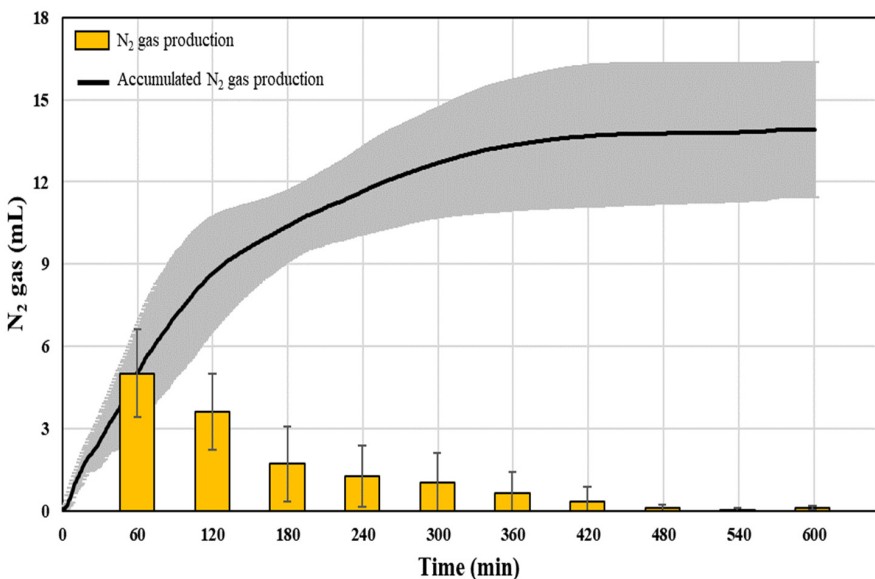

**Figure 2.** Changes in nitrogen gas amount (bar) as a function of time (min) and accumulated nitrogen gas amount (line) during the total reaction time. The grey area indicates the standard deviation for accumulated $N_2$ gas production.

The SAA varied depending on the reaction time, as shown in Table 1. The maximum rSAA (0.449 gN$_2$/gVSS/day) and minimum rSAA (0.111 gN$_2$/gVSS/day) were observed at 60 min and the end of the reaction (600 min), respectively. At the saturation point (360 min), rSAA (0.197 gN$_2$/gVSS/day) was approximately 40% of the maximum rSAA.

**Table 1.** Changes in the rSAA value over the reaction time.

| Time (min) | SAA (gN$_2$/gVSS/Day) |
| :---: | :---: |
| 60 | 0.449 ± 0.133 |
| 120 | 0.384 ± 0.083 |
| 360 | 0.197 ± 0.033 |
| 600 | 0.111 ± 0.022 |

These results showed that the SAA value could be easily calculated from the amount of biogas produced using a respirometer until the reaction was completed without analyzing the nitrogen compounds.

### 3.2. Comparison of rSAA and mSAA According to Temperature and Granule Size

The reliability and accuracy of the SAA analysis method using a respirometer were examined. The SAA was analyzed using two methods (rSAA and mSAA) according to granule size and temperature.

Herein, the two methods showed a similar trend according to granule size and temperature. Both rSAA and mSAA showed the highest SAA values at 35 °C regardless of the granule size, followed by 40 and 30 °C (Figure 3). The temperature range, in which the presence and activity of anammox bacteria were detected, was very wide (−5~80 °C). However, the optimal temperature for most of the anammox species used in the wastewater treatment was between 30 and 40 °C [22]. Therefore, the optimal temperature for anammox bacteria is 35 °C.

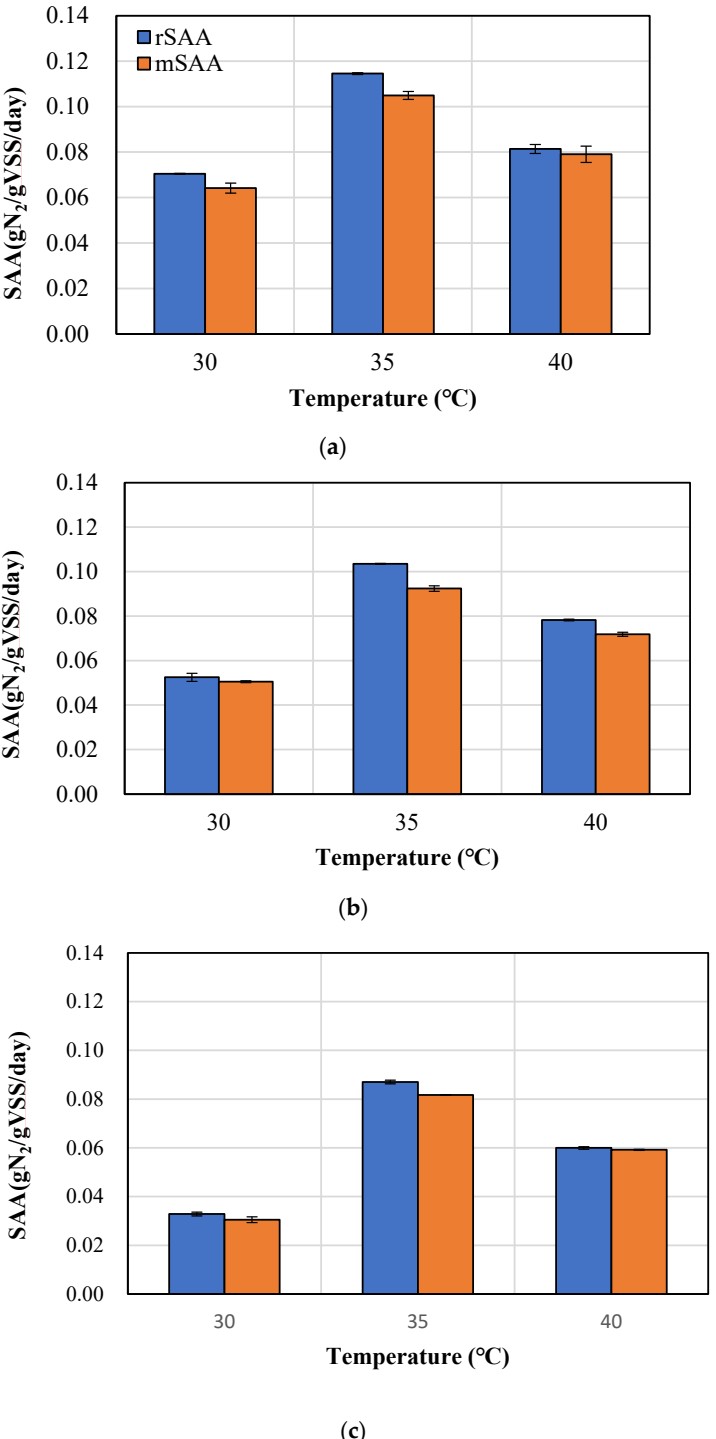

(**a**)

(**b**)

(**c**)

**Figure 3.** The rSAA and mSAA changes according to the temperature by granular size. (**a**) Granule size <1 mm, (**b**) granule size 1~2 mm, and (**c**) granule size >2 mm.

The SAA decreased with the increasing granule size and showed a strong negative linear relationship, according to the granule size under each temperature condition (Figure 4). A granular size of <1 mm showed the highest SAA regardless of the temperature, followed by 1–2 and >2 mm. Anammox bacteria with a small granular sludge are highly active due to the fact that the nutrients are quickly and actively supplied to the center of the granules. However, a granular sludge larger than 2 mm had a low SAA. The larger granule size causes extensive tunnels and empty spaces inside the granule, which lead to the low activity and reduction of granule stability [23].

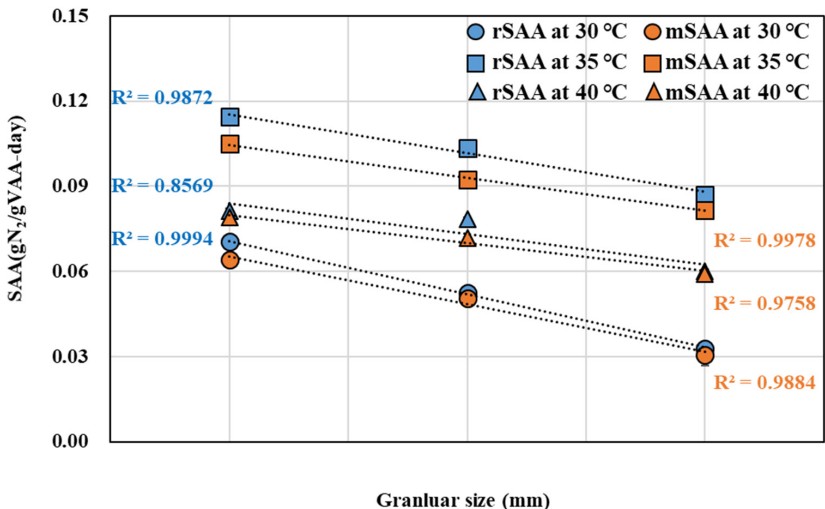

**Figure 4.** The rSAA and mSAA changes according to the granule size by temperature.

The *t*-test and Wilcoxon rank-sum test were performed to determine if the SAA values using the two methods are significantly different. Prior to this step, a Shapiro–Wilk normality test was performed for the SAA value under each condition (Table 2). All of the data appeared to follow a normal distribution since the *p*-values resulting from the Shapiro–Wilk normality test were >0.05. All of the datasets showed a normal distribution. However, the *t*-test and Wilcoxon rank-sum test were performed together since the number of samples was too small (N = 6). At each condition, the *p*-values from the *t*-test and Wilcoxon rank-sum test for rSAA and mSAA were <0.05, which indicates that the SAA values using the two methods were similar. Therefore, a respirometer is well-suited for SAA analysis. However, since there is a limit for the evaluation of the process state, it is necessary to use this value as an indirect indicator, along with various factors that can confirm the process state.

**Table 2.** *p*-values resulting from the Shapiro–Wilk normality, *t*-test, and Wilcoxon rank-sum test for rSAA and mSAA.

| Item | $rSAA_{30°C}$ | $mSAA_{30°C}$ | $rSAA_{35°C}$ | $mSAA_{35°C}$ | $rSAA_{40°C}$ | $mSAA_{40°C}$ |
|---|---|---|---|---|---|---|
| No. [1] | 6 | 6 | 6 | 6 | 6 | 6 |
| N test [2] | 0.276 | 0.402 | 0.232 | 0.430 | 0.051 | 0.487 |
| T test | 0.711 | | 0.223 | | 0.591 | 0.394 |
| W test [3] | | 0.485 | | 0.240 | | |
| **Item** | $rSAA_{<1\ mm}$ | $mSAA_{<1\ mm}$ | $rSAA_{1–2\ mm}$ | $mSAA_{1–2\ mm}$ | $rSAA_{>2\ mm}$ | $mSAA_{>2\ mm}$ |
| No. | 6 | 6 | 6 | 6 | 6 | 6 |
| N test | 0.066 | 0.401 | 0.267 | 0.337 | 0.249 | 0.189 |
| T test | 0.604 | | 0.326 | | 0.601 | 1.000 |
| W test | | 0.485 | | 0.589 | | |

[1] No. of samples; [2] Shapiro–Wilk normality test; [3] Wilcoxon rank-sum test.

## 4. Conclusions

In conclusion, the SAA values from a respirometer (rSAA) showed a similar pattern to the SAA values (mSAA) from the conventional method, according to granule size and temperature. The best SAA value was shown at 35 °C with a granule size of <1 mm. Moreover, the statistical analysis showed no significant differences between rSAA and mSAA. When measuring SAA using the respiratory, the fact that the anammox reaction occurs predominantly should be sufficiently reviewed in advance, since the gas composition cannot be confirmed. Our findings indicate that since the method using the respiratory system can be simply and efficiently measured in the field, it will be a powerful way to indirectly check the performance of the process easily with only the SAA result.

**Author Contributions:** J.Y.; Writing—original draft, visualization, writing—reviewing and editing, Y.K.; methodology, formal analysis, investigation, J.K.; investigation, data curation, S.J.; investigation, data curation, S.P.; Resources, T.L.; supervision, funding acquisition, writing—reviewing and editing. All authors have read and agreed to the published version of the manuscript.

**Funding:** This study was financially supported by the National Research Foundation of Korea (NRF) grant funded by the Korea Government (MIST) (NRF-2018R1D1A1B07046741 and NRF-2021R1A6A1A03039572).

**Institutional Review Board Statement:** Not applicable.

**Informed Consent Statement:** Not applicable.

**Conflicts of Interest:** The authors declare no conflict of interest.

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
