# Peer review of "A Simple Analysis Method of Specific Anammox Activity Using a Respirometer"

_applsci, doi:10.3390/app12031121_

Round 1
Reviewer 1 Report
This manuscript presents a evaluation of a simple method for determining the SAA using a respirometer. Afterwards, they applied this method to investigate the effects of the granule size and the temperature on the SAA. Finally, for the sake of completion, the authors compared these results with other obtained from conventional analysis methods.
From my inspection of the manuscript, I find it suitable for publication in Applied Sciences after improving the manuscript in accordance with my suggestions:
(1) The Anammox process is not a novel process. It was described by Mulder and Graff in 1995 (26 years ago!). (Ref. Mulder, A., Graaf, A.A., Robertson, L.A. y Kuenen, J.G. (1995). Anaerobic ammonium oxidation discovered in a denitrifying fluidized bed reactor. FEMS Microbiology Ecology 16(3):177-184.). Please modify your considerations in this matter.
(2) How could you differentiate the N2 production from the CO2 obtained by the reaction between ammonium and carbonic acid in the respirometers facilities? If this measure is not clear, the results could be wrong and, therefore, the conclusions not acceptable.
Author Response
This manuscript presents a evaluation of a simple method for determining the SAA using a respirometer. Afterwards, they applied this method to investigate the effects of the granule size and the temperature on the SAA. Finally, for the sake of completion, the authors compared these results with other obtained from conventional analysis methods.
- Thank you very much for your valuable comment. It is difficult to directly compare our results with other conventional methods because the experimental condition was not same. Thus, two methods (conventional and respirometry methods) were compared under identical conditions and we could get those results in this manuscript.
From my inspection of the manuscript, I find it suitable for publication in Applied Sciences after improving the manuscript in accordance with my suggestions:
(1) The Anammox process is not a novel process. It was described by Mulder and Graff in 1995 (26 years ago!). (Ref. Mulder, A., Graaf, A.A., Robertson, L.A. y Kuenen, J.G. (1995). Anaerobic ammonium oxidation discovered in a denitrifying fluidized bed reactor. FEMS Microbiology Ecology 16(3):177-184.). Please modify your considerations in this matter.
- Thank you for your comment, the sentence was corrected as following;
‘ Anaerobic ammonium oxidation (anammox) is a biological nitrogen removal process with attractive prospects such as no carbon addition, less aeration, lower greenhouse gas generation, and lower sludge production.’
(2) How could you differentiate the N2 production from the CO2 obtained by the reaction between ammonium and carbonic acid in the respirometers facilities? If this measure is not clear, the results could be wrong and, therefore, the conclusions not acceptable.
- In our experiment, since there was no carbon source, it is theoretically unlikely that CO2 could be produced during the reaction. Nevertheless, as you pointed, it is possible to generate CO2 by the reaction between ammonium and carbonic acid. However, we have confirmed that CO2 is not generated by Gas Chromatography analysis.
Reviewer 2 Report
The method of measuring SAA with a respirometer is not new. However, as the authors of the manuscript rightly noted, this method is practically not mentioned in the scientific literature. In this regard, this publication may be of interest to the relevant reader and can potentially be well cited, since the study of various aspects of the anammox process is still in great demand in the world. The authors need to improve the Materials and Methods section. The main comment is whether this method can be versatile and robust enough? I left a comment on this in the pdf file.

Author Response
- Line 3, What is the principal novelty of this respirometer? this word seems superfluous here
- Thank you for your comment, the word was deleted.
- Line 51, There is a more accurate stoichiometry obtained for planktonic anammox, which is not affected by the mass transfer factor (see 10.1016/j.watres.2014.04.017)
- Thank you for your comment, as you pointed, the stoichiometry in Lotti et al., 2014 was more accurate that the stoichiometry in Strous et al., 1998. However, our purpose is to determine whether a respirometer could be used. Since the experiment was performed according to the stoichiometry of Strous et al., 1998, the stoichiometry was not corrected.
- Line 75. Granules size? Temperature? Initial concentrations of soluble N? Media composition? Replicates?
- In step 1, granule not classified by size were used. The detailed experimental information was added as following;
‘All batch experiments (3 gVSS/L) were performed in duplicate under a thermostatic chamber maintained at 35±â„ƒ except for step 2 (On-lab Co., Busan, Korea). The reactors were operated under complete mixing condition using a stirrer at 120 rpm. The composition of the medium was as follows: 100 mg/L NH4+-N, 100 mg/L NO2--N, 1.010 g/L Na-HCO3, 0.055 g/L KH2PO4, 0.005 g/L CaCl2·5H2O, 0.5 g/L , MgSO4·7H2O, 0.010 g/L FeSO4·7H2O, 0.005 g/L EDTA.’
- Line 98, Have you witnessed the removed NO2--N divided by the removed NH4+- N was < 1.32 in your experiments? Briefly explain why such a coefficient was taken, or provide Reference to works that used the same principle.
- There are several stoichiometric index for ammonium/nitrite (1:1.32 by Strous et al.(1998); 1;1.15 by Lotti et al.(2014); 1:1.31 by Jetten et al.(1999); 1:1.24 by Vanotti et al.(2006)). The stoichiometric index in our experiment was very similar to 1:1.32 by Srous et al.(1998).
- Many studies (Dapena-Mora et al., 2007, Enzyme and Microbial Technology, 40: 859-865; Daverey et al., 2015, International Biodeterioration & Biodegradation, 102: 89-93; Ramos et al., 2015, Chemical Engineering Journal 279: 681-688) followed the stoichiometric index by Strous et al., (1998)
In our experiment, ammonium and nitrite were supplied in ration of 1 but removed in ratio of 1:1.32. So, we used 1.32 as a coefficient based on the well-known stoichiometry in Strous et al., 1998.
- Line 100, subscript
- N2 was corrected with N
- Line 111, Replicates? Mixing? Media composition? Anaerobic conditions
- The detailed experimental information was added as following;
‘All batch experiments (3 gVSS/L) were performed in duplicate under a thermostatic chamber maintained at 35±â„ƒ except for step 2 (On-lab Co., Busan, Korea). The reactors were operated under complete mixing condition using a stirrer at 120 rpm. The composition of the medium was as follows: 100 mg/L NH4+-N, 100 mg/L NO2--N, 1.010 g/L Na-HCO3, 0.055 g/L KH2PO4, 0.005 g/L CaCl2·5H2O, 0.5 g/L , MgSO4·7H2O, 0.010 g/L FeSO4·7H2O, 0.005 g/L EDTA.’
- Line 114, How were the granules fractionated by diameter?
- The granules were separated using a sieve for each size.
- Line 130, Please clarify what the grey area in the diagram means?
- The grey area means standard deviation for accumulated nitrogen gas production.
- Line 132, you should use the same units (see line 103-104)
- Unit was corrected with gN2/gVSS/day.
- Line 136, You used a fairly high concentration of N-NO2, which increases the likelihood of emitting N2 Have you checked the N2O levels in biogas? Have you taken into account the contribution of water vapor to the biogas volume?
- In our experiments, since the stoichiometric index for ammonium and nitrite was similar to 1:1.32 and the reaction occurred in the absence of COD, even if nitrite was high, it was expected that N2O was little generated by heterotrophic denitrification.
- The contribution to water vapor was not included because no change in the liquid phase was observed before and after the reaction.
- Line 136, rephrase the sentence. It looks like you use respirometer for analysing the nitrogen compounds
- The sentence was corrected as following;
- ‘These results showed that the SAA value could be calculated easily from the amount of biogas produced until the reaction was completed using a respirometer without analyzing the nitrogen compounds.’
- Fig 3, What is mS...?
- The graph was corrected
- Line 184, You have not reported the limitations of this method. For example, to what extent this method can be used for SAA measurments in: 1) microbial communities containing anammox bacteria, with a high proportion of heterotrophic denitrifiers (SNAD or similar processes); 2) biomass from industrial anammox-based reactors, where the working concentrations of nitrite are much lower than those that you used in the method; 3) biomass with a low content of anammox bacteria and a high content of heterotrophic microorganisms, which can also emit gases (N2, CO2), for example, due to the degradation of inactive (in your proposed experimental conditions) biomass. How robust and versatile is your proposed method?
- When measuring SAA using the respiratory, it should be sufficiently reviewed in advance that the anammox reaction is predominantly occurring because the gas component cannot be confirmed.
- Since the method using the respiratory system can be quickly and simply measured in the field, it will be a good way to indirectly check the performance of the process easily only with the SAA result.
Reviewer 3 Report
The presented method with the use of a respirometeris very interesting and intriguing - congratulations on the research idea. The manuscript is well written, the individual sections are consistent and the results are graphically and statistically well prepared. I found a few shortcomings in the text, although they did not affect the substantive quality of the manuscript, e.g. different fonts (Ln 145 - Celsius degrees notation), no space between Conclusion and acknowledgments, etc. Therefore, I am asking the authors to read the manuscript carefully and remove all the shortcomings before finally accepting it for publication.
Author Response
- Thank you for your valuable comments, the manuscript has been carefully reviewed.
Round 2
Reviewer 2 Report
Dear Authors, the proposed method is more or less OK, but you need to better discuss the limitations of this method:
- For your tests, you used anammox biomass adapted to fairly high nitrite concentrations. However, anammox-based wastewater treatment reactors with a well-established anammox communities (SNAD, etc.) typically operate at lower working nitrite concentrations, than you used for testing. Are you sure that such a high nitrite concentration (100 mg N-NO2/L) will not suppress the anammox community when testing SAA for "anammox communities adapted to low nitrite concentration" and lead to skewed SAA test results? Please add some discussion on this point.
- You made a completely unconvincing argument about the effect of water vapor on the biogas volume generated during SAA tests ("The contribution to water vapor was not included because no change in the liquid phase was observed before and after the reaction"). As you may know, 1 ml of water produces 1 liter of water vapor. Considering that the volume of generated biogas in your tests was relatively small, then you could simply not notice changes in the volume of liquid, at the same time, the contribution of water vapor could reach 10% of the volume of biogas (for example, this value can be observed for biogas formed in anaerobic digesters). If you only rely on the volume of biogas to indirectly determine the SAA, you should have a better understanding of the biogas composition. Does it consist only of N2? Have you checked the composition to prove it?
In the section "Materials and Methods" you for some reason did not mention that anaerobic conditions for SAA tests should be established, i.e. the headspace in batch reactors must be inert gas. Did you secure this?
I have serious concerns about the Eq. 3. You used factor n, "which is one when the ratio of the removed NO2--N divided by the removed NH4+-N is ≥ 1.32, and 1/1.32 when the ratio of removed NO2--N divided by the removed NH4+-N is < 1.32". I asked you in the first round of revision, why did you use n=1/1.32 when the ratio of removed NO2--N divided by the removed NH4+-N is < 1.32; and your respond was "There are several stoichiometric index for ammonium/nitrite (1:1.32 by Strous et al.(1998); 1;1.15 by Lotti et al.(2014); 1:1.31 by Jetten et al.(1999); 1:1.24 by Vanotti et al.(2006)). The stoichiometric index in our experiment was very similar to 1:1.32 by Srous et al.(1998)".
1. It follows from this that, firstly, the stoichiometric index strongly depends on the experimental conditions for determining SAA, and secondly, on the culture of anammox used (planktonic/granules, anammox genera). This point should also be taken into account in the calculations.
2. Also, the following is not clear: for example, the ratio of removed NO2 - N divided by the removed NH4 + -N is little less than 1.32 (for example, 1.31). Then we have to use n = 1/1.32, which immediately reduces Δ??2 by 32%. In your opinion, is this acceptable ?? So I repeat my question: Have you witnessed the removed NO2--N divided by the removed NH4+- N was < 1.32 in your experiments? Briefly explain why such a coefficient was taken, or provide Reference to works that used the same principle.
Round 3
Reviewer 2 Report
Dear Authors,
Please be more careful, it seems to me that in equations 3 and 4 you have mixed up "?â„Ž? ??????? ??4 + −?" and "?â„Ž? ??????? ??2 −− ?".
General comment. It is clear that the purpose of the work was to show the fundamental possibility of using an alternative method for determining SAA using a respirometer. The authors fully succeeded in this, congratulations. But this method cannot be simply copied and used to study other anammox communities, because, for example, at used N-NO2 concentrations in real anammox-based wastewater treatment installations, SAA values ​​can be skewed due to inhibition. If I were the author of this work, I would have mentioned this and other possible restrictions either in the discussion or in the conclusion of the article. So that someone does not get discouraged when trying to repeat this method.
